# Characteristics of leaf nutrient resorption efficiency in Tibetan alpine permafrost ecosystems

Guibiao Yang[1,2], Meifeng Deng [1,2], Lulu Guo [1,2,3], Enzai Du [4,5], Zhihu Zheng [1,2,3], Yunfeng Peng [1,2], Chunbao Zhao[1,2,3], Lingli Liu [1,2,3] & Yuanhe Yang [1,2,3] ✉

Nutrient resorption is an important strategy for nutrient conservation, especially in permafrost ecosystems where plant growth is limited by nutrients. Based on the measurements mainly derived from tropical, subtropical and temperate regions, current projections suggest that resorption efficiency is higher for leaf nitrogen (N) than for phosphorus (P) in cold regions. However, these projections have not been fully validated due to the lack of observations in permafrost ecosystems. Here, we carry out a large-scale sampling campaign along a permafrost transect on the Tibetan Plateau. Our results show that, in contrast with the prevailing view, resorption efficiency is higher for leaf P than N in permafrost ecosystems (75.1 ± 1.8% $vs.$ 58.7 ± 1.5%; mean ± standard error). Our results also reveal that leaf P resorption efficiency is higher in permafrost ecosystems than in global herbaceous plants, while there is no difference for leaf N resorption efficiency. Interestingly, there is a trade-off between leaf N resorption efficiency and soil N mineralization rate, but no such pattern exists for P. These results illustrate the unique characteristics of plant nutrient resorption in permafrost ecosystems and advance our understanding of nutrient conservation strategies in little-studied permafrost regions.

Net primary production, the main pathway for carbon (C) uptake in terrestrial ecosystems, is extensively limited by essential nutrients, especially the elements nitrogen (N) and phosphorus (P)[1–5]. Nutrient resorption, defined as the transportation of nutrients from senesced leaves to other plant tissues, plays a crucial role in maintaining nutrient cycling within ecosystems since it determines the rate at which nutrients are returned from plant material to the soil[6,7]. This process can reduce the dependence of plants on the external nutrient supply[8] and has been assumed to be one of the most important nutrient conservation strategies[9,10], playing a vital role in vegetation C sequestration. It has been reported that leaf nutrient resorption can contribute up to 31% of the total plant nutrient demand for N and 40% for P in

terrestrial ecosystems[11]. Therefore, a better understanding and a more precise estimate of leaf nutrient resorption are essential for accurately evaluating plant nutrient status and predicting terrestrial C dynamics under changing environmental conditions.

Given its critical role in mediating plant growth and vegetation C sequestration, leaf nutrient resorption has received considerable attention[1,12–15], and a number of observational studies have been conducted across tropical, subtropical, and temperate regions[1,16,17]. However, the more pristine regions, such as the permafrost regions—which occupy around 15% of the global land area[18], store ~30% of the global surface soil organic C[19] and are considered to be substantial atmospheric C sources under future climate warming[20,21]—have

[1]State·Key Laboratory of Vegetation and Environmental Change, Institute of Botany, Chinese Academy of Sciences, Beijing, China. [2]China National Botanical Garden, Beijing, China. [3]University of Chinese Academy of Sciences, Beijing, China. [4]State Key Laboratory of Earth Surface Processes and Resource Ecology, Faculty of Geographical Science, Beijing Normal University, Beijing, China. [5]School of Natural Resources, Faculty of Geographical Science, Beijing Normal University, Beijing, China. ✉e-mail: yhyang@ibcas.ac.cn

received less attention. Global-scale extrapolation based on current observations shows that leaf P resorption efficiency is larger than that of N in tropical, subtropical, and temperate regions at low altitudes, while leaf N resorption efficiency is larger than that of P in permafrost regions at high altitudes and latitudes[1,5]. However, a small number of field studies are inconsistent with this global projection, observing a stronger plant P limitation than that of N in permafrost ecosystems[22–24]. Given that leaf nutrient resorption efficiency depends on soil nutrient supply and is higher in nutrient-limited regions[1,14,25], lower soil P supply and stronger plant P limitation may be associated with the larger leaf P resorption efficiency than that of N in permafrost ecosystems[1,26,27]. Nevertheless, due to the limited number of direct measurements, the relative sizes of leaf N and P resorption efficiency in permafrost zones across a broad geographical scale remain unclear.

To address this knowledge gap, we carry out a large-scale sampling campaign in the largest alpine permafrost region around the world—the Tibetan Plateau[28]. With ~1.06 × 10⁶ km² of the plateau (40%) being underlain by permafrost[29], it is an ideal environment for exploring the spatial pattern of leaf nutrient resorption efficiency in permafrost ecosystems (Fig. 1a). We conduct a large-scale sampling campaign along a permafrost transect of 1100 km across the Tibetan Plateau during the peak growing season and wilting period in 2021, and sample aboveground vegetation at 30 sites (Fig. 1b). We determine mature and senesced leaf N and P concentrations, and calculate leaf N and P resorption efficiencies to explore the large-scale pattern of leaf nutrient resorption across this study area. We also synthesize published studies to explore the differences in leaf nutrient resorption efficiencies in herbaceous plants between Tibetan permafrost ecosystems and global terrestrial ecosystems. Our results demonstrate that, contrary to current projections, the leaf resorption efficiency of P is higher than that of N. Mean leaf N resorption efficiency is approximately equivalent to the corresponding value for global herbs, while mean P resorption efficiency is at the upper end of the range for global herbs. To explore the potential relationship between plant nutrient

resorption and soil nutrient supply, we further measure in situ topsoil N and P mineralization rates at 30 sites across the Tibetan permafrost region. The results reveal that there is a trade-off between leaf N resorption efficiency and soil N mineralization rate, but no such pattern exists for P.

## Results and discussion
### Comparison of leaf P and N resorption efficiency across Tibetan alpine permafrost ecosystems

Our results showed that leaf nutrient resorption efficiencies exhibited large spatial variability across the study area (Fig. 2; Supplementary Table 1). Leaf N resorption efficiency ranged from 37.9% to 72.3%, while leaf P resorption efficiency varied from 44.4% to 87.3%. Leaf P resorption efficiency was found to be significantly higher than that of N across Tibetan permafrost ecosystems (75.1 ± 1.8% vs. 58.7 ± 1.5%; hereafter, values are expressed as mean ± standard error; degrees of freedom (df) = 59, $P < 0.001$, cohen's $d = 1.8$, 95% confidence interval (CI) = 13.9–18.9; Fig. 2a), in contrast to the prevailing view that leaf N resorption efficiency was larger than leaf P resorption efficiency in cold regions[1].

The higher plant P resorption efficiency relative to that of N across the Tibetan permafrost region may be attributable to either plant resource utilization strategies or a more severe restriction of P supply relative to N. Consider the first case: inorganic P is more easily adsorbed onto soil particles than inorganic N[30]. To mobilize these insoluble soil P compounds, plants need to release organic compounds (such as carboxylates and phosphatases) through their roots[31,32], making soil insoluble P accessible for plant uptake[33–35]. This process requires larger amounts of energy for P absorption per unit than for N absorption. However, in the low-temperature Tibetan permafrost ecosystems, there is less energy available for plants to mobilize soil insoluble P[36,37]. Consequently, plants must intercept more P from other pathways, such as increasing leaf P resorption efficiency. The second possible explanation for the higher plant P resorption efficiency relative to that of N is that the pattern may be due to the more severe restriction of P

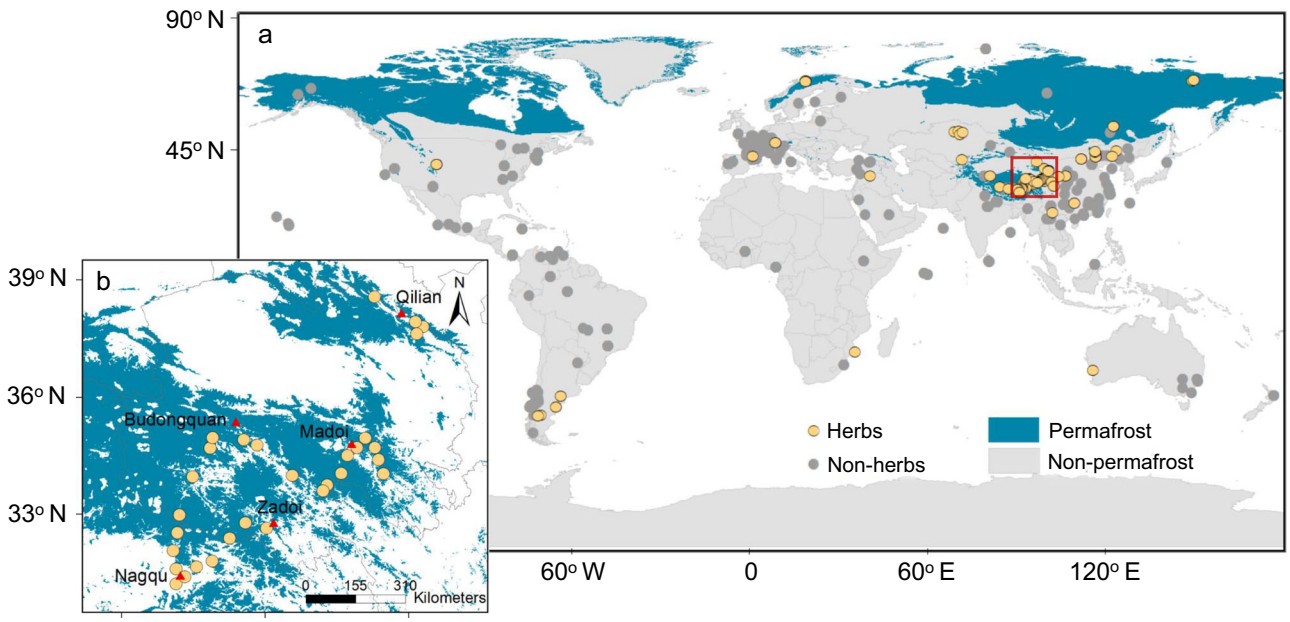

**Fig. 1 | Geographic distributions of study sites. a** The distribution of non-herbs (gray dots) and herbs (orange dots) in the global database. The global database was compiled from published datasets (see details in Supplementary Note 4). **b** The layout of sampling sites across the Tibetan alpine permafrost region. The map was created using ArcMap 10.7 (Environmental Systems Research Institute, Inc.,

Redlands, CA, USA) based on data derived from the National Snow & Ice Data Center[64] (https://nsidc.org/data/ggd318/versions/2)/(https://nsidc.org/about/data-use-and-copyright) and Zou et al.[29] (https://tc.copernicus.org/articles/11/2527/2017/)/CC BY (https://creativecommons.org/licenses/by/3.0/), respectively.

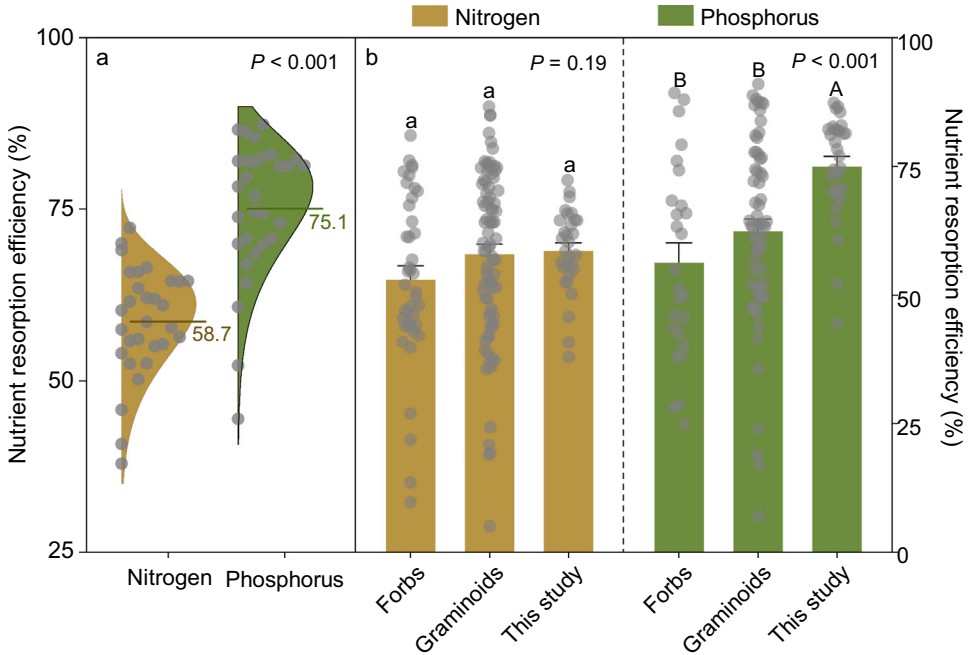

**Fig. 2 | Concentration-based leaf nutrient resorption efficiencies in plants among Tibetan alpine grasslands, global forbs, and graminoids. a** Assessing the differences in leaf N and P resorption efficiencies in plants across Tibetan alpine grasslands with a paired-samples *t*-test (two-sided, *n* = 30). **b** Comparative analysis of leaf nutrient resorption efficiencies between plants across Tibetan alpine grasslands and global forbs and graminoids with independent-samples *t*-tests (two-sided). The global database was compiled from published studies (see details in Supplementary Note 4). Sample sizes of leaf N and P resorption efficiencies are 83 and 65 for graminoids, 43 and 27 for forbs in global datasets, respectively. Data are represented as the means ± SE (standard error). *** (unadjusted *P* < 0.001), and different letters (unadjusted *P* < 0.05) represent significant differences (lowercase letters for N and capital letters for P).

supply relative to N. Plant nutrient conservation strategies depend on soil nutrient supply, with plants exhibiting greater nutrient resorption efficiency in nutrient-poor environments[14,27]. Consistent with this deduction, the soil gross rate of N mineralization was significantly higher than that of microbial N immobilization (df = 47, *P* < 0.001, cohen's *d* = 0.6, CI = 0.6–0.9)[38], while the soil gross rate of P mineralization was lower than that of microbial P immobilization despite there being no statistical significance across our study area (df = 9, *P* = 0.17, cohen's *d* = 0.4, CI = −0.8 to 0.3; Supplementary Fig. 1). These results demonstrate that there is a more severe restriction of P supply across our study area, which could therefore be responsible for the higher leaf P resorption efficiency observed in this study.

**Comparison of leaf P resorption efficiency in herbaceous plants between Tibetan permafrost ecosystems and global terrestrial ecosystems**

Our results indicated that mean leaf N resorption efficiency over the Tibetan alpine permafrost region was approximately equivalent to that in global forbs (df = 72, *P* = 0.09, cohen's *d* = 0.4, CI = −0.8 to 12.0) and graminoids (df = 112, *P* = 0.01, cohen's *d* = 0.2, CI = −5.6 to 7.0), while P resorption efficiency was remarkably higher than that in global forbs (df = 56, *P* < 0.001, cohen's *d* = 1.3, CI = 10.9–26.5) and graminoids (df = 94, *P* < 0.001, cohen's *d* = 1.0, CI = 5.6–19.7; Fig. 2b). Consistently, the mass-based leaf P resorption efficiency was at the upper end of global forbs and graminoids (see details in Supplementary Note 1). In addition, both mean leaf N and P resorption efficiencies across the Tibetan alpine permafrost region were higher than the corresponding global average (Supplementary Fig. 2). These comparisons revealed that, across this poorly-studied permafrost region, plants had a high leaf P resorption efficiency. Further analysis revealed that P concentrations were low in the senesced leaves, varying from 0.2 to 0.9 g kg⁻¹, with a mean value of 0.4 ± 0.03 g kg⁻¹ (Supplementary Table 1). According to the criterion of Killingbeck[6], if plants thoroughly resorb nutrients, P concentrations in the senesced leaves would be

below the complete resorption boundary (<0.4 g kg⁻¹ for P). The senesced leaf P concentrations observed in our study were close to the complete resorption boundary at 23 of the 30 sites, suggesting that leaf P resorption proficiency was high across Tibetan permafrost ecosystems. Taken together, these two lines of evidence support the view that, in this little-studied permafrost region, plants have a high P conservation capacity from senesced leaves.

The high values of leaf P resorption efficiency across the Tibetan permafrost region could be ascribed to the following two aspects. First, such a pattern may result from the cold climate in permafrost ecosystems. It has been reported that plants can economically regulate the balance between plant nutrient resorption and nutrient uptake from soils to meet their nutrient demands[39]. Of them, root P absorption is an energy-dependent active process, which doesn't allow plants to take up P efficiently from soils at high energetic costs under cold conditions[36,37]. As a consequence, low temperatures can impede plant P absorption from soils[20], leading to high leaf P resorption efficiency in permafrost ecosystems. Second, the high leaf P resorption efficiency may be due to the severe restriction of plant-available P in the soil. It is widely accepted that available P is governed by the release of inorganic P by weathering of primary minerals and the transfer of exchangeable P pools (secondary mineral P, occluded P, and organic P)[33]. However, the unique environmental conditions—marked by low temperature and air oxygen concentration—would inhibit rock P weathering and the transformation of various P fractions to plant-available P forms[40,41], despite the high weatherable rock P concentrations in the geologically-young and cold ecosystems[42]. This situation may thus result in a severe restriction of available P for plants and the observed high leaf P resorption efficiency across Tibetan permafrost ecosystems.

**Plant nutrient resorption strategies across Tibetan alpine permafrost ecosystems**

Three basic strategies, nutrient concentration control, stoichiometry control, and nutrient limitation control, are often considered when

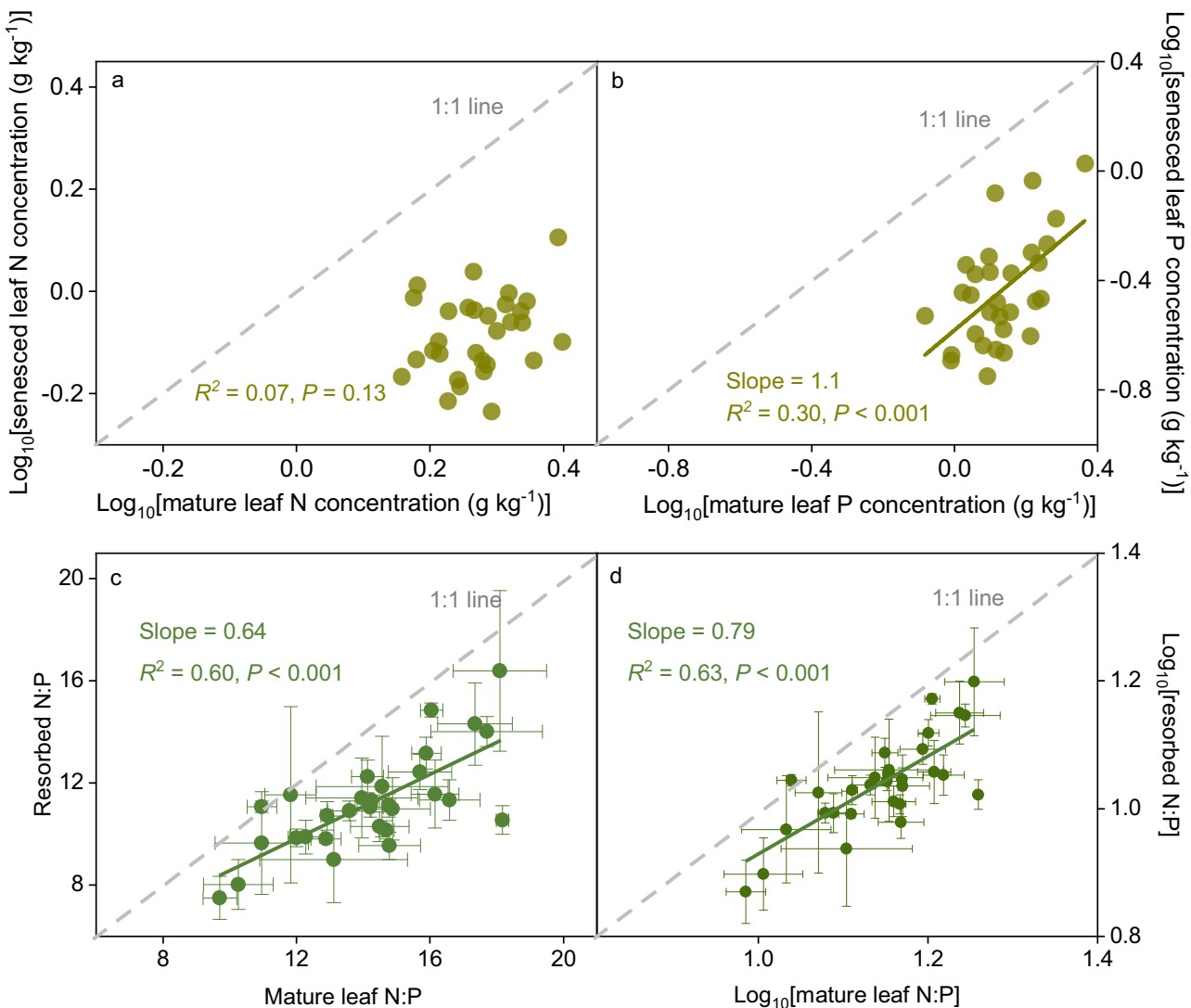

**Fig. 3 | Assessment of three control strategies underlying plant nutrient resorption across Tibetan alpine grasslands. a** Relationship between N concentrations in the mature and senescent leaves. **b** Association between P concentrations in the mature and senescent leaves. **c** Linkage between resorbed N:P and leaf N:P concentrations in the mature leaves. **d** Correlation between $\log_{10}$-transformed resorbed N:P and $\log_{10}$-transformed leaf N:P concentrations in the mature leaves. The gray dashed line in each panel is the 1:1 line. Only significant relationships are shown using solid lines. Error bars denote SE of mean at each site ($n = 3$). Statistics (slope, $R^2$, and $P$ value) are shown for the linear mixed-effects models with two-sided $t$-tests (unadjusted $P < 0.05$).

exploring the regulation of plant nutrient resorption[43–45]. As to the nutrient concentration control strategy, according to Kobe[25], a linear regression is widely used to characterize the relationship between $\log_{10}$-transformed nutrient concentrations in senesced and mature leaves, with the slope reflecting the mutual relationship between leaf nutrient resorption efficiency and mature leaf nutrient concentration. A slope greater than 1 indicates that nutrient resorption is more efficient in mature leaves with low nutrient concentrations (see details in "Methods" section)[45]. To evaluate the potential role of the nutrient concentration control strategy in regulating nutrient resorption across our study area, we analyzed the relationships between nutrient concentrations in senesced and mature leaves (Fig. 3a, b; Supplementary Fig. 3). Our results revealed that, after $\log_{10}$-transformation, the senesced leaf N concentration didn't show any significant relationship with the mature leaf N concentration ($P = 0.13$; Fig. 3a). However, the $\log_{10}$-transformed senesced leaf P concentration was positively correlated with the $\log_{10}$-transformed mature leaf P concentration ($P < 0.001$), with the slope being greater than 1 ($P = 0.048$; Fig. 3b), indicating that the strategy of nutrient concentration control does not regulate leaf N resorption but plays an important role in mediating leaf P resorption.

Under the stoichiometry control strategy, plants resorb nutrients according to the living leaf nutrient stoichiometry since nutrients are transported from the senesced to living leaves along the flow of solutes[46]. Due to this point, resorbed N and resorbed P should be positively associated, with their ratio proportional to the concentration ratio of N to P in the mature leaf (see details in "Methods" section)[12,44,47]. To test this strategy on plant nutrient resorption across our study area, we explored the relationship between leaf N and P resorption efficiencies across the Tibetan alpine permafrost region. Our results showed that there was a positive correlation between them (Supplementary Fig. 4), indicating that N and P could be resorbed based on their stoichiometry in the mature leaf. We then assessed the relationship between the resorbed nutrient ratio and the nutrient concentration ratio of N to P in the mature leaf, and found that the ratio of resorbed N to resorbed P was proportional to the concentration ratio of N to P in the mature leaf (Fig. 3c). This result indicated that plants resorbed nutrients from senescing leaves in a proportion that paralleled the original nutrient stoichiometry in the mature leaves and proved the existence of the stoichiometry control on plant nutrient resorption across the Tibetan alpine permafrost region[26].

Regarding the nutrient limitation control strategy, plant resorbs nutrient at a rate dependent upon each nutrient's limitation statu: the limiting nutrient can be more resorbed by the plant than other nutrients[13,27,48]. Under this framework, a linear regression is usually developed to characterize the relationship between the $\log_{10}$-transformed resorbed nutrient ratio and the nutrient concentration ratio of N to P in the mature leaf, with the slope reflecting the mutual relationship of the leaf N to P resorption efficiency ratio with the concentration ratio of N to P in the mature leaf[25,43,45]. A slope greater than 1 indicates that nutrient resorption increases when nutrient limitation is aggravated (see details in "Methods" section). To test the nutrient limitation control strategy across our study area, we compared the slope of the regression between the $\log_{10}$-transformed resorbed nutrient ratio and the mature leaf nutrient concentration ratio of N to P with the 1:1 line. The slope of this regression was 0.79, significantly less than 1 ($P < 0.001$; Fig. 3d), suggesting that the resorbed N was dependent upon P. This comparison confirmed that plants exhibited the strategy of nutrient limitation control on nutrient resorption[45]. Overall, our results illustrated that the three basic strategies existed simultaneously in the permafrost region on the Tibetan Plateau.

### Relationships between leaf nutrient resorption efficiency and soil mineralization rate

It is generally assumed that a trade-off exists between leaf nutrient resorption efficiency and soil nutrient supply[14]. To explore their potential relationships, we measured in situ soil N and P mineralization rates at 30 sites across the Tibetan alpine permafrost region. Our results showed that topsoil N and P mineralization rates exhibited large spatial variability, ranging from 125.1 to 475.7 ng cm$^{-2}$ d$^{-1}$ and 0.1 to 1.8 ng cm$^{-2}$ d$^{-1}$, with means of $232.5 \pm 15.5$ and $0.8 \pm 0.07$ ng cm$^{-2}$ d$^{-1}$, respectively (Supplementary Table 1). More importantly, leaf N resorption efficiency was significantly correlated with soil N mineralization rate (Fig. 4a), but there was no clear association between leaf P resorption efficiency and soil P mineralization rate (Fig. 4b). We also observed insignificant relationships between leaf N to P resorption efficiency ratio and soil N, P mineralization rates. However, there was a significant negative relationship between leaf N to P resorption efficiency ratio and soil N to P mineralization rate (Supplementary Fig. 5).

These results demonstrated that there was a trade-off between N resorption efficiency and mineralization rate, but no such pattern occurred between leaf P resorption efficiency and soil P mineralization rate.

Leaf nutrient resorption and soil nutrient mineralization are both important pathways for the acquisition of nutrients[14]. Increasing soil N mineralization enhances the amount of plant-available N in the soil, potentially reducing the plants' reliance on leaf N resorption[14,33]. Such a trade-off would likely yield a negative relationship between leaf N resorption efficiency and soil N mineralization rate. An interesting question that arises is why there is no trade-off between leaf P resorption efficiency and soil P mineralization rate? The absence of such a pattern can likely be ascribed to the unique strategy of plant P acquisition. Unlike inorganic N forms ($NH_4^+$-N and $NO_3^-$-N) which can be easily taken up by plants following microbial mineralization, less than 20% of inorganic P is mobile in the soil and directly available for plants to absorb, because of the rapid chemical fixation and slow diffusion rates[39,49]. In such a case, plants have to evolve diverse strategies to obtain soil P[33,39]. In the Tibetan permafrost ecosystems, the dominant species, including *Carex moorcroftii*, *Kobresia pygmaea* and *K. humilis*, and *K. tibetica*, have developed special structures, such as cluster roots[50], which have the function of mining soil immobile inorganic P[32]. As a result, P acquisition by plants can be less reliant on mobile inorganic P derived from soil mineralization, leading to no trade-off relationship between leaf P resorption efficiency and soil mineralization rate as observed here.

In summary, based on a large-scale sampling campaign at 30 sites along a 1100-km permafrost transect on the Tibetan Plateau, we explored the geographical pattern of leaf nutrient resorption efficiency across this poorly-studied permafrost region. Our results revealed that leaf P resorption efficiency was higher than that of N, implying the possible existence of P limitation on plant growth in this study area. This finding challenges the prevailing view that N is the primary limiting nutrient in permafrost ecosystems[1,5]. Our results also showed that there was a trade-off between leaf N resorption efficiency and soil N mineralization rate, but no such trade-off for P: a difference between the N and P cycles within ecosystems that should be taken into account when exploring how nutrient cycles regulate plant C uptake in this little-studied permafrost region.

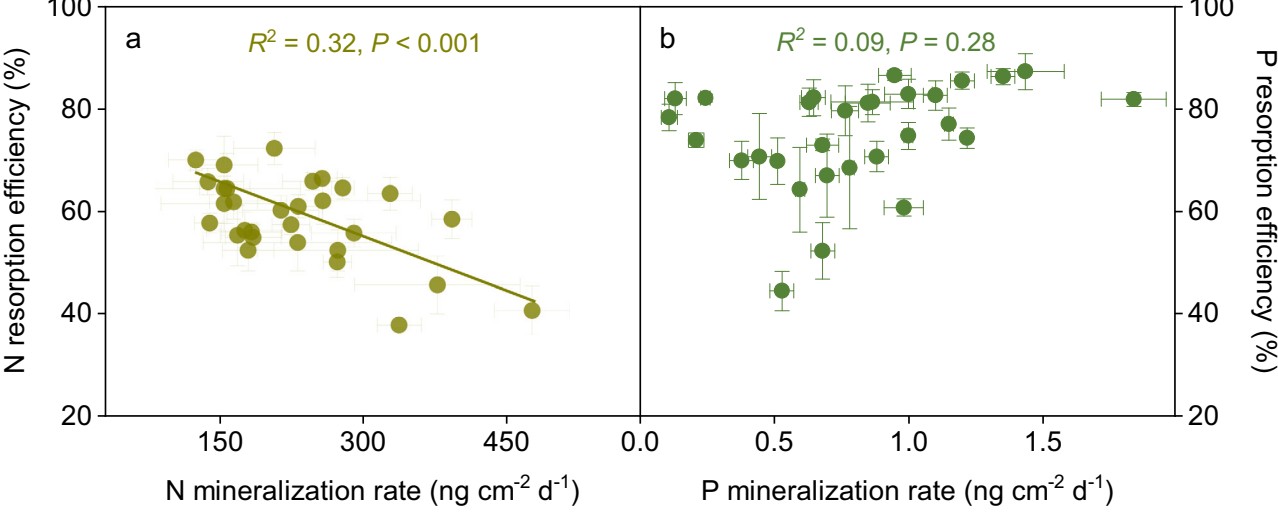

**Fig. 4 | Trade-off between plant nutrient resorption and soil nutrient supply across Tibetan alpine grasslands. a** Relationship of leaf N resorption efficiency with topsoil N mineralization rate. **b** Association of leaf P resorption efficiency with topsoil P mineralization rate. A significant relationship is shown by a solid line. Error bars denote SE of mean at each site ($n = 3$). Statistics ($R^2$ and $P$ value) are shown for the generalized linear mixed-effects models with two-sided $t$-tests (unadjusted $P < 0.05$).

## Methods

### Study area

This study was conducted across the permafrost region of the Tibetan Plateau (Fig. 1b). The plateau is the world's largest area of alpine permafrost in the low to middle latitudes ($-1.1 \times 10^6$ km²) with an average elevation of 4000 m above sea level[28]. It makes up about 75% of the alpine permafrost in the Northern Hemisphere[29]. The permafrost across the study area is mainly categorized as discontinuous and sporadic types[18], with a current average active-layer thickness of -1.9 m, ranging from 0.9 to 3.2 m[51]. Climate, soil properties, and vegetation types exhibit large spatial variations across the region. Mean annual temperature ranges between −2.9 and 7.0 °C, and mean annual precipitation varies from 129 to 590 mm. Alpine steppe, alpine meadow, and swamp meadow are the dominant vegetation types, with the corresponding dominant species being *Stipa purpurea* and *C. moorcroftii*, *K. pygmaea* and *K. humilis*, and *K. tibetica*, respectively. There is limited anthropogenic nutrient input in the region, with rates of atmospheric N and P deposition being <1 g N m⁻² yr⁻¹ and 0.1 g P m⁻² yr⁻¹, respectively[52,53].

### Large-scale field sampling

To explore the basic characteristics of community-level plant nutrient resorption across the permafrost region, we created a unique dataset based on a large-scale sampling campaign at 30 sites along a 1100 km permafrost transect. Field campaigns were carried out during the peak growing season from mid-July to mid-August and in the wilting period from early to late October in 2021. Three criteria were considered when selecting the 30 sampling sites. First, the sites should be representative of the three main permafrost regions across the study area. To ensure this was the case, 10 sites were located in the Madoi section on the eastern plateau, 15 sites in the Budongquan-Nagqu-Zadoi section in the central part of the plateau, and 5 sites in the Qilian section on the northeastern plateau (Fig. 1b). Second, different vegetation types should be covered across the 30 sites. To meet this criterion, 5 sites were located in alpine steppe, 13 in alpine meadow, and 12 in swamp meadow (Supplementary Table 2). Third, the 30 sites should represent broad environmental gradients. For example, across the chosen 30 sites, soil organic C ranged from 8.0 to 222.0 g kg⁻¹, and total N content varied from 0.9 to 17.1 g kg⁻¹ (Supplementary Table 2).

At each site, three individual quadrats (50 × 50 cm²) were established as replicates along the diagonal line of a 10 × 10 m² plot where vegetation was characterized. Within each quadrat, all living (mature) and recently senesced (naturally) but still attached leaves were sampled during the peak growing season and the wilting period, respectively. To minimize the potential bias caused by varying plant characteristics between the two periods, the field sampling campaigns at each site at the two times were carried out by the same participants, and the senesced leaf quadrats were located adjacent to mature leaf quadrats (Supplementary Fig. 6). All collected samples of mature and senesced leaves were oven-dried at 65 °C to constant weight and roughly ground with a crusher. A portion of them was then finely ground with a ball mill before chemical analyses.

### Calculation of leaf nutrient resorption efficiency

Community-level mature and senesced leaf N concentrations were determined by an elemental analyzer (Vario EL III, Elementar, Germany). Leaf P concentrations were measured with a spectrophotometer (ICAP6300, Thermo Fisher Scientific, Waltham, MA, USA) after a microwave-assisted digestion with $H_2SO_4$ and $H_2O_2$ at 380 °C for -3 h. Before the measurement of leaf N and P concentrations, in-house standards (Phenylalanine) were used to calibrate the concentration of total N, and a standard calibration curve for P concentrations was prepared using a serial dilution of inorganic P solution. During the measurement of leaf N and P concentrations, reference materials (GBW10020 GSB-11 Citrus leaves, approved by General

Administration of Quality Supervision, Inspection and Quarantine of the People's Republic of China) were run with the samples to further evaluate the effect of calibration and to check the accuracy of N and P concentrations. The analytical accuracies of the elemental analyzer and spectrophotometer were better than 0.1% for N concentration and 1 ppb for P concentration, respectively.

Based on mature and senesced leaf nutrient concentrations, leaf nutrient resorption efficiency was calculated from Eq. (1)[6,25]:

$$NuRE = \frac{Nu_{mat} - Nu_{sen}}{Nu_{mat}} \times 100\% \tag{1}$$

where $Nu_{mat}$ and $Nu_{sen}$ are the mature and senesced leaf N concentrations (g kg⁻¹), respectively. The abbreviation, NuRE, stands for leaf nutrient resorption efficiency. Plant absolute nutrient resorption was calculated by the difference in nutrient concentrations in the mature and senesced leaves[6,25]. Notably, although the two sampling times coincided approximately with the peak and end of plant growth on the Tibetan Plateau[54], plant nutrient resorption might be continuing beyond the sampling periods[55], leading to the fact that leaf nutrient resorption efficiency could be affected by any potential seasonal variations. It is suggested that in future work, more attention should be paid to such potential seasonal variations in order to generate a more comprehensive estimate of plant nutrient resorption across this study region.

### Identification of three control strategies underlying plant nutrient resorption

Three basic strategies for mediating plant nutrient resorption, i.e., the nutrient concentration control strategy, the stoichiometry control strategy, and the nutrient limitation control strategy, were explored. To verify the presence of the nutrient concentration control strategy, we analyzed the relationships between $\log_{10}$-transformed senesced and mature leaf nutrient concentrations. Specifically, a conceptual model (Eq. 2), proposed by Kobe[25], determines the relationship between senesced and mature leaf nutrient concentrations:

$$Nu_{sen} = a \times Nu_{mat}{}^{b} \tag{2}$$

After converting with $\log_{10}$ transformation, a linear regression is shown between $\log_{10}(Nu_{sen})$ and $\log_{10}(Nu_{mat})$ as:

$$\text{Log}_{10}(Nu_{sen}) = \text{Log}_{10}(a) + b \times \text{Log}_{10}(Nu_{mat}) \tag{3}$$

By combining Eqs. (1) and (3), leaf nutrient resorption efficiency can be expressed as:

$$NuRE = \left(1 - a \times Nu_{mat}{}^{b-1}\right) \times 100\% \tag{4}$$

Thus, if $b > 1$, nutrient resorption efficiency decreases with increasing mature leaf nutrient concentration (i.e., nutrient concentration control). In contrast, if $b \leq 1$, plant nutrient resorption does not adhere to the nutrient concentration control strategy[25,45].

To explore the stoichiometry control strategy, we evaluated the relationships between leaf N and P resorption efficiencies, and between the resorbed nutrient ratio and nutrient concentration ratio of N to P in the mature leaf. In the case when both correlations are positive, plants resorb nutrients in a proportion that parallels the original nutrient stoichiometry in mature leaves. In the other cases, plant nutrient resorption does not follow the stoichiometry control strategy[25,45].

To examine the nutrient limitation control strategy, we analyzed the relationship between the $\log_{10}$-transformed resorbed nutrient ratio of two elements (resorbed $Nu_1{:}Nu_2$) and the $\log_{10}$-transformed concentration ratio of two elements in mature leaves (leaf $Nu_1{:}Nu_2$).

Specifically, a power law regression (Eq. 5) is developed to characterize the relationship between the resorbed $Nu_1:Nu_2$ and leaf $Nu_1:Nu_2$ according to Kobe[25]:

$$\text{Resorbed } Nu_1 : Nu_2 = \varepsilon \times (\text{Leaf } Nu_1 : Nu_2)^{\lambda} \quad (5)$$

By converting with a $\log_{10}$ transformation, a linear regression between $\log_{10}$ (Resorbed $Nu_1:Nu_2$) and $\log_{10}$ (Leaf $Nu_1:Nu_2$) is shown as Eq. (6):

$$\text{Log}_{10}\left(\text{Resorbed } Nu_1 : Nu_2\right) = \text{Log}_{10}(\varepsilon) + \lambda \times \text{Log}_{10}\left(\text{Leaf } Nu_1 : Nu_2\right) \quad (6)$$

By combining Eqs. (1) and (5), $Nu_1RE:Nu_2RE$ can be characterized as:

$$Nu_1RE : Nu_2RE = \varepsilon \times (\text{Leaf } Nu_1 : Nu_2)^{\lambda-1} \quad (7)$$

Leaf $Nu_1:Nu_2$ is widely represented as an indicator of plant nutrient limitation status. Leaf $Nu_1:Nu_2$ increases when plant's $Nu_2$ limitation is aggravated. Thus, $\lambda < 1$, $Nu_1RE:Nu_2RE$ increases when $Nu_2$ limitation is aggravated, which accords with the nutrient limitation control strategy; if $\lambda \geq 1$, plant nutrient resorption does not follow the nutrient limitation control strategy[25,45].

## In situ measurements of soil N and P mineralization

In situ topsoil N and P mineralization rates were determined with field incubation of intact soil cores using ion-exchange resin bags[22,56]. Specifically, three plastic tubes (5 cm in diameter) were inserted to a depth of 10 cm near the individual quadrats at each site. The use of three plastic tubes adopted in the current study was based on classic literature[57] and the fact that, on the Tibetan Plateau, alpine grasslands are relatively homogeneous (see details in Supplementary Note 2). All aboveground vegetation within the tubes was clipped, the tubes pulled out, and their bottoms covered with nylon mesh bags containing ~5 grams mixed-bed ion-exchange resin (Sigma Amberlite 150 mixed-bed resins). Before the installation, these resin bags were shaken in 2 M KCl solution for 3 h to saturate exchange sites with $K^+$ and $Cl^-$ ions, washed with distilled water, and dried at 60 °C. The tubes, now each covered with a resin bag, were then put back in their original locations, and soils at the same depth were sampled near each tube. The resin bags were set out in the field from mid-June to mid-July and removed from mid-September to mid-October 2021. During the incubation period, the bags were replaced once a month to avoid oversaturation of the ion-exchange resin. On removal, the resin bags were rinsed with deionized water.

After incubation, soil samples from within the tubes were collected. Soil samples and resin bags were both transferred to the laboratory (State·Key Laboratory of Vegetation and Environmental Change, Institute of Botany, Chinese Academy of Sciences) and stored at −20 °C. In the laboratory, they were extracted for 30 min with a 1 M KCl solution at room temperature. The extracted liquids were analyzed for $NH_4^+$-N, $NO_3^-$-N concentrations using a flow injection analyzer (Autoanalyzer 3 SEAL; Bran and Luebbe, Norderstedt, Germany). The inorganic P concentrations in the extracted liquids were analyzed using the vanado-molybdate method with a spectrophotometer (UV-2550; Shimadzu, Kyoto, Japan). Finally, the soil N or P mineralization rate was determined by calculating the difference between the post-incubated amount of inorganic N ($NH_4^+$-N and $NO_3^-$-N) or inorganic P in the soil and resin bags with the corresponding values in the soil before incubation. These rates were expressed on the basis of the bottom area of the plastic tube and the incubation time (ng $cm^{-2}$ $d^{-1}$).

To ensure the accuracy of the soil N and P mineralization rates, we adopted the following three steps during the measurement period. First, all samples, including before- and post-incubated soils and resin bags, were kept frozen at −20 °C during transportation and storage to prevent the conversion of inorganic N or P[58]. Second, inorganic N or P concentrations of before-incubated soils were measured simultaneously with those of the post-incubated soils and resin bags to reduce systematic errors across different measurement times. Third, during the measurement of soil inorganic N and P concentrations, in-house standards (quantitative inorganic N and P solutions) were run with the samples to check the accuracy of the N and P concentrations.

## Global data synthesis

A global dataset was compiled from published datasets before October 2024 by searching the Web of Science (https://webofscience.clarivate.cn). The key words "leaf nitrogen resorption" or "leaf phosphorus resorption" were used to search published studies using the following criteria: (1) the selected studies should report at least one of leaf N and P resorption efficiencies; (2) nutrient resorption efficiency should be directly presented or indirectly calculated by nutrient concentrations in mature and senesced leaves; (3) litter samples should be collected from newly withered leaves rather than from the decomposed litter when determining nutrient concentrations in senesced leaves; (4) data should be obtained from forest, shrub or grassland ecosystems.

To ensure the consistency of data exclusion across all sources, we first checked the "Site Description" section and identified the specific ecosystem for all potential studies considered for inclusion. We then excluded any data from managed ecosystems, such as urban forests, agroforests, croplands, sown pastures, and fertilized plantations. We also excluded data from wetland and aquatic ecosystems, such as mangroves, salt marshes, riparian wetlands, rivers, lakes, ponds, and reservoirs. In addition, where manipulative experiments were performed, we didn't use data from the treatment conditions. Besides, all literature cited in the previous meta-analysis by Du et al.[1] were included in this dataset. The final dataset consists of 134 studies containing 998 observations of leaf N resorption efficiencies and 913 observations for leaf P resorption efficiencies. Of these observations, 83 and 65 are for N and P resorption efficiencies, respectively, in graminoids, while 43 and 27 are for N and P resorption efficiencies, respectively, in forbs.

To compare plant nutrient resorption between Tibetan alpine grasslands and global herbs as well as the whole global average, we calculated global leaf nutrient resorption efficiencies on a concentration basis using Eq. (1). If the mass-based leaf nutrient resorption efficiency was originally estimated ($NRE_m$), as done in earlier literature[1,43], the mass loss correction factor (MLCF) was used to calculate concentration-based leaf nutrient resorption efficiency by means of Eq. (8):

$$NuRE = (1 - (1 - NuRE_m) \times MLFC) \times 100\% \quad (8)$$

where MLCF is 0.780 for evergreen broadleaves, 0.784 for deciduous broadleaves, 0.745 for conifers, 0.640 for forbs, and 0.713 for graminoids, respectively.

## Statistical analyses

We tested for the normality and homoscedasticity of all data (using Levene's test) before analysis. A series of statistical analyses were conducted to explore the basic characteristics of leaf nutrient resorption efficiency across Tibetan alpine permafrost ecosystems. Specifically, a paired-samples $t$-test was performed to compare the means of leaf N and P resorption efficiencies at 30 sites across these ecosystems. Independent-samples $t$-tests were employed to compare the differences in leaf nutrient resorption efficiency across Tibetan alpine grasslands and the corresponding averages from global herbs. The sample sizes for the independent-samples $t$-tests were considered to be adequate based on the fact that all power values in these analyses exceeded 0.8 (see details in Supplementary Note 3)[59].

Given the lack of independence among observations within one site, linear mixed-effects models were used to explore strategies underlying plant nutrient resorption across the Tibetan alpine grasslands. In these models, the dependent variables were treated as the fixed factors, and the replicates nested within the site were treated as the random factors. Generalized linear mixed-effects models were employed to explore relationships between leaf nutrient resorption efficiency and soil mineralization rate, with leaf nutrient resorption efficiencies as the fixed factors and the replicates nested with site as the random factors. Significance was determined at both $P < 0.05$ and cohen's $d > 0.2$. Linear and generalized linear mixed-effects models were analyzed using the "lme4" package in R software 4.3.1[60–62].

### Reporting summary

Further information on research design is available in the Nature Portfolio Reporting Summary linked to this article.

### Data availability

The experimental measurements produced in this study have been deposited in the Figshare data repository (https://doi.org/10.6084/m9.figshare.28103306.v1)[63]. A complete list of sources for the global dataset is provided in Supplementary Note 4. The corresponding datasets are available from the corresponding author upon request.

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

## Acknowledgements

We thank all the scientists who contribute to the global database used in this study. We also appreciate Dr. Yuxuan Bai (Institute of Botany, Chinese Academy of Sciences) and Dr. Jinsheng Li (College of Resources and Environment, Anhui Agricultural University) for the assistance on statistical analyses, and Dr. Dianye Zhang (Institute of Botany, Chinese Academy of Sciences) and Dr. Hao Chen (State Key Laboratory of Biocontrol, School of Ecology, Sun Yat-Sen University) for their help on results interpretation. Permissions to work and collect samples across the study area were granted by the Three-River-Source National Park Management Bureau. This work was supported by the National Key Research and Development Program of China (2022YFF0801901, Y.Y.) and the National Natural Science Foundation of China (32425004, Y.Y., and 32201359, G.Y.).

## Author contributions

Y.Y. and G.Y. designed this research. G.Y. performed field samplings. G.Y., Z.Z., and C.Z. performed laboratory experiments. E.D., G.Y., M.D., L.G., and L.L. performed global data synthesis. G.Y. analyzed data. G.Y., Y.P., and Y.Y. wrote the manuscript with input from other co-authors.

## Competing interests

The authors declare no competing interests.
