## [Peer Review file · Nature Communications]

Characteristics of leaf nutrient resorption efficiency in Tibetan alpine permafrost ecosystems

Corresponding Author: Professor Yuanhe Yang

Version 0:

Reviewer comments:

Reviewer #1

(Remarks to the Author)

Overall, I found the paper to be highly engaging. It reports that the mean plant nitrogen (N) and phosphorus (P) resorption rates in the Tibetan alpine ecosystem are at the upper end of the global synthesis range, a noteworthy finding given the limited studies on plant P resorption in this region. This work significantly enhances our understanding of plant nutrient dynamics under climate change, where ecosystem functions and carbon cycling might be substantially affected.

However, the paper requires some clarification in the methods and additional editing, especially minor grammatical corrections for consistency and readability. While the analysis presented is appropriate, more details should be included in the global data synthesis section. Additionally, the rationale for choosing linear mixed-effects models and the sample size should be clearly explained.

Regarding the finding that plant P resorption is significantly higher than N resorption, it would be beneficial to include a more thorough discussion of the potential reasons for this result to ensure confidence in the findings, especially since your results are being upscaled to a global context.

Reviewer #2

(Remarks to the Author)

Yang and colleagues conducted a sampling campaign along a 1,100 km transect across Tibetan alpine permafrost grassland to explore the N and P resorption efficiency of the grass at the community level. They reported that the mean plant N and P resorption (efficiency), i.e., NRE and PRE, were at the upper end of global synthesis range, and that plant P resorption (efficiency) is higher than N resorption efficiency (75.1%P vs. 58.7%N), which suggested a severe P limitation relative to N on plant growth in the Tibetan alpine permafrost grassland.

Their field work is very impressive considering the arduousness of fieldwork on the Qinghai Tibet Plateau. I believe the conclusion about the N and P nutrient status is generally correct. Especially they used data from the in-situ measurements of soil nutrient mineralization to further validate that these permafrost grasslands were more limited by P rather than N. However, the major results are not surprising. Herbaceous plants usually show higher N and P resorption efficiency than woody plants. According to the global results reported by Vergutz et al (2012) (which the authors have cited), the N and P resorption efficiency of herbaceous plants (globally) were respectively 67.5~73.3% (NRE) and 79.0~83.3% (PRE) for graminoid, which is the major plant functional type in Tibetan Plateau; or respectively 61.1~73.4% (NRE) and 64.0~77.4% (PRE) for global forbs. All the results of global herbs are higher than Yang's results in this study, and PRE > NRE for global herbs also indicated that herbaceous plants are more limited by P than N. Actually, this is one of the major conclusions in our current study too. However, it is strangely that the authors compared their NRE and PRE of herbaceous plants on the Qinghai Tibet Plateau in this paper with those of all plant species worldwide (Figure 2). This comparison is clearly unreasonable, leading the author to the wrong conclusion: the NRE and PRE of herbaceous plants in permafrost on the Qinghai Tibet Plateau were higher than the global average.

Moreover, the NRE and PRE were OVERESTIMATED in Yang's study because they did not compensate for the leaf mass loss, caused by the strong winds frequently occurs on Tibetan Plateau. Considering this overestimation, their results are even lower than the global results for herbaceous plants reported by Vergutz et al (2012).

Version 1:

Reviewer comments:

Reviewer #1

(Remarks to the Author)

This is my second time reviewing this manuscript, and I thoroughly enjoyed reading it. I am impressed by the substantial improvements made to the approach and the robustness of the analyses. The authors have done an excellent job, and this study provides valuable insights into how climate change may influence the dynamics and functions of permafrost—an ecosystem of critical importance that remains understudied.

The introduction has been significantly improved, and I am pleased with the clarity and logical flow of the language. The authors have addressed my previous questions and comments comprehensively, and I am satisfied with their thoughtful and thorough responses.

Reviewer #2

(Remarks to the Author)

The authors have made major revisions according to the previous comments. The quality of the paper has been greatly improved. I have no further suggestions, and recommend accepting this manuscript for publication in NC.

Responses to Reviewer #1

[Comment 1] *Overall, I found the paper to be highly engaging. It reports that the mean plant nitrogen (N) and phosphorus (P) resorption rates in the Tibetan alpine ecosystem are at the upper end of the global synthesis range, a noteworthy finding given the limited studies on plant P resorption in this region. This work significantly enhances our understanding of plant nutrient dynamics under climate change, where ecosystem functions and carbon cycling might be substantially affected.*

[Response] We are very grateful to the reviewer for the positive and insightful comments on our manuscript! These comments listed below help us to conduct a thorough revision on this manuscript. We really appreciate this professional review which greatly improved our paper. Thank you! Detailed modifications please see our responses to the following comments.

Major comments:

[Comment 2] *However, the paper requires some clarification in the methods and additional editing, especially Minor grammatical corrections for consistency and readability. While the analysis presented is appropriate, more details should be included in the global data synthesis section. Additionally, the rationale for choosing linear mixed-effects models and the sample size should be clearly explained.*

[Response] Following the reviewer's comments, we made the following major changes in the revised MS:

- **We added more descriptions to clarify our method.** Particularly, the details about the experimental design (*Comments 12 and 15; Page 13, lines 300-310 and Page 17, lines 402-404*) and steps taken to ensure the data accuracy (*Comments 13 and 16; Page 14, lines 329-338 and Pages 18-19, lines 430-439*) were provided in the revised MS.
- **We added the detailed descriptions of global data synthesis,** involving the criteria for selection and exclusion of data from the published studies (*Comment 17; (Page 19, lines 445-451 and Page 19, lines 455-460)*).
- **We explained the rationality for choosing linear mixed-effects models**

(*Comment 20*). In our study, the 30 sampling sites were selected along a 1,100-km transect, and three individual quadrats were established at each site. Given the lack of independence among observations within one site, linear mixed-effects models are ideal for such situations in which the assumption of independence in traditional models is violated when modeling the relationship between variables (Bates *et al.*, 2014) (Page 21, lines 491-495).

- **We conducted power analyses to ensure the adequacy of sample size for the independent-samples *t* tests** (*Comment 21*). The results showed that all power values in these analyses exceed 0.8. A power value of 0.8 is usually used on the basis of the ratio of Type II (β) to Type I error (α) (Cohen, 1988). This to say, sample sizes were adequate for these independent-samples *t* tests (Page 21, lines 485-487).
- **We asked a native English-language speaker (Alistair Culf, Email: a.culf@hotmail.com) for language check** through the whole manuscript to reduce grammatical corrections for consistency and readability.

Overall, we really appreciate these professional and constructive comments which greatly improved our paper. See detailed responses in the following comments.

[Comment 3] Regarding the finding that plant P resorption is significantly higher than N resorption, it would be beneficial to include a more thorough discussion of the potential reasons for this result to ensure confidence in the findings, especially since your results are being upscaled to a global context.

[Response] Very good comment! Following the reviewer's suggestion, we provided two potential reasons for the finding that plant P resorption is significantly higher than N resorption. **The higher plant P resorption efficiency relative to that of N across the Tibetan alpine permafrost region may be attributable to either plant resource utilization strategies or a more severe restriction of P supply relative to N. Consider the first case:** inorganic P is more easily adsorbed onto soil particles than N inorganic (Parfitt, 1979). To mobilize these insoluble soil P compounds, plants need release organic compounds (such as carboxylates and phosphatases) through their roots (Sabine *et al.*, 2017; Raven *et al.*, 2018), making soil insoluble P accessible

for plant uptake (Lambers *et al.*, 2008; Hans *et al.*, 2019; Wen *et al.*, 2021). This process requires larger amounts of energy for P absorption per unit than for N absorption. However, in the low-temperature Tibetan permafrost ecosystems, there is less energy available for plants to mobilize soil insoluble P (Yan *et al.*, 2016; Dolezal *et al.*, 2021). Consequently, plants must intercept more P from other pathways, such as increasing the leaf P resorption efficiency. **The second possible explanation is the more severe restriction of P supply relative to N.** Plant nutrient conservation strategies depend on soil nutrient supply, with plants exhibiting greater nutrient resorption efficiency in nutrient poor environments (Güsewell, 2010; Deng *et al.*, 2018). Consistent with this deduction, the soil gross rate of N mineralization was significantly higher than that of microbial N immobilization ($P < 0.001$) (Mao *et al.*, 2020), while the soil gross rate of P mineralization was lower than that of microbial P immobilization despite there being no statistical significance across our study area ($P = 0.17$; Supplementary Fig. 1; unpublished data). These results demonstrate that there is a more severe restriction of P supply across our study area, which could therefore be responsible for the higher leaf P resorption efficiency observed in this study. We have mentioned these points in the revised MS (Pages 5-6, lines 110-131).

Figure R1. Ratio of soil gross nutrient mineralization to microbial nutrient immobilization across the Tibetan alpine permafrost region. The transformation rates of N and P are reanalyzed from Mao *et al.* (Mao *et al.*, 2020) and Ziliang Li (unpublished data), and determined using the ^{15}N tracer technique (Hart *et al.*, 1994)

and the ^{33}P tracer technique (Wanek, 2019), respectively. Red line indicates the value 1. Data are represented as the means \pm SE. *** indicates a significant difference from 1 (two-sided t -tests; $P < 0.001$); ns, insignificant difference.

Minor comments:

*[Comment 4] **Ln 1**:* The title is inappropriate; "adaptation" is not an appropriate term. I suggest "alpine permafrost ecosystem."

***Ln 26-29**:* I suggest changing it to "projections hold that larger plant nitrogen (N) resorption relative to phosphorus (P)" to make it more natural.

***Ln 28**:* What's the cold region? It should be "permafrost ecosystem."

***Ln 30**:* "Two times sampling campaign" should be simplified to "two sampling campaigns."

[Response] Done as suggested.

*[Comment 5] **Ln 32-33**:* P resorption rate? What's the statistical confidence? Or what's the standard error? And why significant? What's the p -value?

[Response] Sorry for this confusion. This is plant P resorption efficiency rather than P resorption rate. Standard errors of leaf N and P resorption efficiencies are 1.5 and 1.8%, respectively, and p -value of their significant differences < 0.001 . We have replaced this phrase with 'P resorption efficiency' (Page 2, lines 35-36) and shown standard errors of leaf P and N resorption efficiencies in the revised MS (Page 2, line 34-35). Nevertheless, considering the word limit, p -value was not represented in the section of *Abstract*, but shown in the section of *Results and Discussion* in the revised MS (Page 5, lines 105-107). Thanks for your understanding!

*[Comment 6] **Ln 32**:* Changing "is significantly higher relative" to "was significantly higher relative to" maintains tense consistency.

***Ln 45-47**:* Not clear; suggest changing to "Therefore, a better understanding and more precise estimation of plant nutrient resorption are essential for accurately evaluating plant nutrient status and predicting terrestrial C dynamics at regional and

global scales."

***Ln 45-47**:* Using "approximately" and "around" instead of "~" is more formal and professional in the introduction section.

***Ln 59-61**:* Adding "have" clarifies the tense. "Long-standing paradigm" is more accurate than "long-term standing paradigm."

***Ln 62-63**:* "Suggested" is more commonly used in scientific literature than "advocated." Changing "at nutrient-limited region" to "in nutrient-limited regions" corrects the preposition and pluralizes "region" for consistency.

[Response] Done as suggested.

[Comment 7] ***Ln 94**:* Capitalizing key words in the title aligns with standard scientific publication formats. I suggest "High Plant Nutrient Resorption in Alpine Ecosystem across Tibetan Plateau."

***Ln 95-96**:* "Based on" is more concise than "On the basis of." The phrase "geographical patterns" is clearer and more precise.

***Ln 99-100**:* Simplifying the sentence structure improves readability and flow. I suggest "Plant N resorption efficiency ranged from 37.9% to 72.3%, averaging $58.7 \pm 1.5\%$, while plant P resorption efficiency varied from 44.4% to 87.3%, with a mean value of $75.1 \pm 1.8\%$."

***Ln 101-102**:* Removing "of them" and "respectively" simplifies the sentence without losing meaning.

***Ln 109**:* "Ulteriorly" is not a commonly used word and can be replaced with "subsequently" for clarity. Correct the spelling of "varying" and "respectively."

[Response] Done as suggested.

[Comment 8] ***Ln 112-113**:* "Nutrient resorption by the plant" is more concise than "resorption of nutrient by the plant."

***Ln 120-121**:* Removing "plant" for conciseness because "N and P resorption efficiencies" clearly refers to plant nutrient resorption.

****Ln 127-128****: Adding "than N resorption efficiency" clarifies the comparison being made.

****Ln 129-130****: Splitting the sentence and clarifying the conditions for better readability and grammar. I suggest "Generally, plants have roughly balanced N and P resorption efficiency under balanced growth conditions. However, if there is N or P limitation, plants will resorb a greater proportion of the limited nutrient from senesced leaves."

[Response] Done as suggested.

[Comment 9] ****Ln 134-144****: Adjust the sentence for clarity and ease of understanding. Correct "in-suit" to "in situ" and adjust the preposition for better clarity.

[Response] Following this reviewer's comments, we adjusted the sentence as follows: 'Leaf P resorption efficiency was found to be significantly higher than that of N across Tibetan permafrost ecosystems ($75.1 \pm 1.8\%$ vs. $58.7 \pm 1.5\%$; hereafter, values are expressed as mean \pm standard error; $P < 0.001$; Fig. 2a), in contrast to the prevailing view that leaf N resorption efficiency was larger than leaf P resorption efficiency in cold regions (Du et al., 2020).' (Page 5, lines 104-108).

We also corrected "in-suit" to "in situ" and adjusted the preposition as follows:

'To explore their potential relationships, we measured in situ soil N and P mineralization rates at 30 sites across the Tibetan alpine permafrost region. Our results showed that topsoil N and P mineralization rates exhibited large spatial variability, ranging from 125.1 to 475.7 $\text{ng cm}^{-2} \text{d}^{-1}$ and 0.1 to 1.8 $\text{ng cm}^{-2} \text{d}^{-1}$, with means of 232.5 ± 15.5 and $0.8 \pm 0.07 \text{ ng cm}^{-2} \text{d}^{-1}$, respectively (Supplementary Table 1).' (Page 10, lines 233-238).

[Comment 10] ****Ln 146-148****: Break the sentence into two for clarity and add "However" to clearly indicate the contrast.

****Ln 155-157****: Remove "Actually" for conciseness and ensure consistent tense.

****Ln 158-160****: "Overall" is a more formal transition than "All in all." "Challenge"

is more appropriate than "are against," and "long-standing paradigm" is clearer.

***Ln 167-169**:* Split the sentence for clarity and flow, and remove unnecessary words for conciseness. I suggest "Our results revealed that senesced leaf N concentrations did not show any significant relationship with mature leaf N concentrations ($P = 0.13$; Figure 3a). However, senesced leaf P concentration was significantly correlated with mature leaf P concentration ($P < 0.001$), with a slope of 1.1 (Figure 3b)."

***Ln 181-184**:* Correct "Plant may" to "Plants may" for subject-verb agreement, and simplify for clarity and conciseness.

[Response] Done as suggested.

[Comment 11] *The summary section provides a clear conclusion of the study. However, several areas need refinement for clarity, coherence, and conciseness, such as adding "campaigns" for grammatical accuracy and "a" before "poorly-studied" for clarity in Ln 200, replacing "blind-spot" with "understudied" for a more formal tone, and simplifying the sentence for clarity in Ln 202. Change "evidences" to "evidence" for correct usage, and simplify the sentence for clarity in Ln 212.*

***Ln 238-239**:* Simplify the sentence for clarity and conciseness. Change "two times' field campaigns" to "two field campaigns."

[Response] Done as suggested.

[Comment 12] ***Ln 238-239**:* What were the specific criteria for selecting the 30 sampling sites? Were there any particular environmental or vegetation characteristics considered?

[Response] Three criteria were considered when selecting the 30 sampling sites. **Frist, 30 sites were located in three representative permafrost regions across the study area.** Of them, 10 sites were located in the Madoi section on the eastern plateau, 15 sites in the Budongquan-Nagqu-Zadoi section in the central part of the plateau, and 5 sites in the Qilian section on the northeastern plateau (Yang *et al.*, 2023). **Second,**

different vegetation types were covered across the 30 sites. Alpine grassland including alpine steppe, alpine meadow and swamp meadow are the dominant vegetation types. Of the 30 sites, 5 sites were in alpine steppe, 13 sites in alpine meadow and 12 sites in swamp meadow (Table R1). Third, broad environmental gradients were represented across 30 sites. For example, soil organic carbon and total nitrogen content ranged from 8.0 to 222.0 g kg⁻¹, and from 0.9 to 17.1 g kg⁻¹, respectively (Table R1). We have clearly stated these points in the revised MS (Page 13, lines 300-310).

Table R1. Environmental and vegetation characteristics at 30 sampling sites on the Tibetan Plateau.

N	Lat (°N)	Long (°E)	Altitude (m)	pH	SOC (g kg ⁻¹)	TN (g kg ⁻¹)	IN (mg kg ⁻¹)	IP (mg kg ⁻¹)	Grassland types
1	34.5	99.2	4321	7.6	99.0	8.0	35.0	11.1	SM
2	34.1	99.3	4405	6.3	130.9	9.6	45.0	12.0	AM
3	34.8	99.0	4633	7.4	103.6	8.6	44.2	10.6	SM
4	35.1	98.7	4419	7.2	94.1	7.8	32.1	13.9	AM
5	34.8	98.4	4219	9.1	9.7	0.9	3.7	7.2	AS
6	34.6	98.0	4181	8.2	51.5	4.1	30.6	11.4	AS
7	34.2	97.8	4576	6.9	138.8	10.6	101.9	22.7	AM
8	33.9	97.3	4508	7.6	33.1	2.6	18.1	15.8	AM
9	34.2	96.0	4653	7.5	115.6	8.6	122.7	8.5	AM
10	33.7	97.1	4437	6.8	195.9	14.6	145.2	11.7	AM
11	34.9	94.7	4396	8.0	113.8	8.3	96.1	16.1	SM
12	32.8	95.0	4498	7.1	155.2	11.3	72.0	27.3	SM
13	32.9	94.3	4768	7.4	117.1	8.9	42.7	21.5	AM
14	32.5	93.7	4746	7.4	96.7	7.9	44.6	26.5	AM
15	31.9	93.1	4439	6.9	123.3	8.9	37.6	23.8	AM
16	31.3	91.9	4497	7.8	127.9	9.2	92.2	22.3	SM
17	31.5	92.2	4581	7.6	115.6	8.5	76.8	22.3	SM

18	31.7	91.8	4620	7.5	162.6	12.5	84.2	31.1	SM
19	32.6	91.9	5014	8.7	5.7	0.5	10.4	8.1	AM
20	33.1	91.9	4886	8.5	23.0	1.8	44.2	16.2	AS
21	34.1	92.3	4725	8.8	22.7	1.7	18.5	5.7	AM
22	34.8	92.9	4628	8.8	14.3	1.2	16.6	5.7	AM
23	35.1	94.2	4406	9.5	10.4	1.0	14.6	6.5	SM
24	32.2	91.7	4806	8.8	8.0	0.9	12.3	8.4	AS
25	31.8	92.6	4689	6.9	116.4	8.9	102.1	12.9	AS
26	37.5	100.3	3820	8.1	82.8	7.1	29.1	11.6	SM
27	38.7	99.3	3443	6.5	171.5	14.1	107.0	10.4	SM
28	37.8	101.1	3600	8.5	50.6	4.4	30.9	12.6	SM
29	38.0	100.8	3279	6.4	222.0	17.1	151.6	9.4	AM
30	37.7	100.8	3618	8.4	15.8	14.8	17.1	9.0	SM

Note: N represents sampling site number; AS, alpine steppe; AM, alpine meadow; SM, swamp meadow; Lat, latitude; Long, longitude; SOC, soil organic carbon; TN, total nitrogen; IN, inorganic nitrogen; IP, inorganic phosphorus.

[Comment 13] ***Ln 254-256***: Could you provide more details on the calibration and accuracy of the elemental analyzer and spectrophotometer used in the study?

[Response] Following this comment, we added details on the calibration and accuracy of the elemental analyzer and spectrophotometer as follows: *‘Before the measurement of leaf N and P concentrations, in-house standards (Phenylalanine) were used to calibrate the concentration of total N, and a standard calibration curve for P concentrations was prepared using a serial dilution of inorganic P solution. During the measurement of leaf N and P concentrations, reference materials (GBW10020 GSB-II Citrus leaves, approved by General Administration of Quality Supervision, Inspection and Quarantine of the People's Republic of China) were run with the samples to further evaluate the effect of calibration and to check the accuracy of N and P concentrations. The analytical accuracies of the elemental*

analyzer and spectrophotometer were better than 0.1% for N concentration and 1 ppb for P concentration, respectively. We have clearly stated these points in the revised MS (Page 14, lines 329-338).

[Comment 14] Was any consideration given to potential seasonal variations in nutrient resorption rates beyond the specified sampling periods? If so, how were they addressed?

[Response] Very good comment! We agree with the reviewer nutrient resorption may be going on beyond the sampling periods. Nevertheless, it's challenging to monitor the seasonal variation in nutrient resorption efficiency over the 1,100 km permafrost transect, due to the rugged environment, such as extremely low temperature, oxygen limitation, and traffic inconvenience on the Tibetan Plateau. **We have addressed this point in the revised MS as follows:** *‘Notably, although the two sampling times coincided approximately with the peak and end of plant growth on the Tibetan Plateau (Shen et al., 2022), plant nutrient resorption might be continuing beyond the sampling periods (Lin et al., 2010), leading to the fact that leaf nutrient resorption efficiency could be affected by any potential seasonal variations. It is suggested that in future work more attention should be paid to such potential seasonal variations in order to generate a more comprehensive estimate of plant nutrient resorption across this study region.’* (Page 15, lines 346-352). Thanks for your understanding!

*[Comment 15] **Ln 273-275**:* *Why were three plastic tubes chosen per site, and how does this number ensure the representativeness of the data?*

[Response] Good comment! **Number of repetitions adopted in current study were based on classic literature.** Among them, the most representative one is A.C. Risch et al. (2019) Nature Communications that explore the pattern of field soil net N mineralization across 30 grasslands worldwide using the same method (Risch et al., 2019). **To evaluate the representativeness of the data obtained from three duplicates, we arranged ten plots as 10 replicates within an area of 50 × 50 m² at Gangca county, Qinghai Province, China** (37° 18' N, 100° 15' E; 3,280 m

above sea level) and collected soil samples at a depth of 10 cm in each plot. Then, we measured soil organic C and total P contents, and characterized their means and standard deviations. Based on their means and standard deviations, we randomly selected three samples 100 times. We observed that, for soil organic C, the average values of the three samples across the 100 iterations consistently fell within $\pm 5\%$ of the overall mean (Figure R2a). In the case of total P content, 95 out of the 100 iterations were found to be within $\pm 5\%$ of the mean (Figure R2b). **These results demonstrate that 3 replicates can ensure the representativeness of the related measurements across these grassland ecosystems.** We have clearly stated this point in the revised MS. (Page 17, lines 402-404).

Figure R2. Means of three samples across 100 iterations for soil organic C (a) and total P content (b). The black lines and shades represent the means with positive and negative 5% intervals.

[Comment 16] ***Ln 293***: How were the initial corresponding values in soils before incubation measured, and what steps were taken to ensure their accuracy?

[Response] Initial corresponding values in soils before incubation were measured using the same procedure as the post-incubated amount of inorganic N or P in soils.

Three steps were taken to ensure their accuracy. **First,** all samples including before- and post-incubated soils and resin bags were kept frozen at -20°C during transportation and storage to prevent the conversion of inorganic N or P (Bremner & Black, 1965). **Second,** inorganic N or P concentrations of before-incubated soils were measured simultaneously with those of the post-incubated soils and resin bags to reduce systematic errors across different measurement times. **Third,** during the measurement of inorganic N and P concentrations, in-house standards (quantitative inorganic N and P solution) were run with the samples to check the accuracy of the N and P concentrations. We have clearly stated these points in the revised MS (Pages 18-19, lines 430-439).

*[Comment 17] **Line 299-300**:* Could you specify the criteria used to select included in the global data synthesis? Were there any exclusion criteria?

[Response] Following the reviewer's comment, **we have clearly stated selection criteria in the revised MS as follows:** ' (1) the selected studies should report at least one of leaf N and P resorption efficiencies; (2) nutrient resorption efficiency should be directly presented or indirectly calculated by nutrient concentrations in mature and senesced leaves; (3) litter samples should be collected from newly withered leaves rather than from the decomposed litter when determining nutrient concentrations in senesced leaves; (4) data should be obtained from forest, shrub or grassland ecosystems.' (Page 19, lines 445-451).

We have also clearly stated exclusion criteria in the revised MS as follows: 'We then excluded any data from managed ecosystems, such as urban forests, agroforest, croplands, sown pastures and fertilized plantations. We also excluded data from wetland and aquatic ecosystems, such as mangroves salt marshes, riparian wetlands, rivers, lakes, ponds and reservoirs. In addition, where manipulative experiments were performed, we didn't use data from the treatment conditions.' (Page 19, lines 455-460).

*[Comment 18] **Line 303**:* How did you ensure the exclusion of data from wetlands, aquatic, and agricultural ecosystems was consistent across all sources?

[Response] Sorry for this non-detailed description. To ensure the consistency of data exclusion across all sources, we checked ‘Site Description’ section and identified the specific ecosystem for all potential study considered for inclusion.

We have clearly stated this point in the revised MS as follows: ‘To ensure the consistency of data exclusion across all sources, we first checked the ‘Site Description’ section and identified the specific ecosystem for all potential study considered for inclusion. We then excluded any data from managed ecosystems, such as urban forests, agroforest, croplands, sown pastures and fertilized plantations. We also excluded data from wetland and aquatic ecosystems, such as mangroves salt marshes, riparian wetlands, rivers, lakes, ponds and reservoirs. In addition, where manipulative experiments were performed, we didn’t use data from the treatment conditions.’ (Page 19, lines 453-460).

*[Comment 19] **Line 305-307**:* Could you provide more detail on the equations used to calculate global plant N and P resorption efficiencies? Are these equations widely accepted in the field, or were any modifications made?

[Response] Following this comment, we have provided the detail on the equations used to calculate global leaf N and P resorption efficiencies as follows: ‘To compare plant nutrient resorption between Tibetan alpine grasslands and global herbs as well as the whole global average, we calculated global leaf nutrient resorption efficiencies on a concentration basis using eq. 1.

$$\text{NuRE} = (\text{Nu}_{\text{mat}} - \text{Nu}_{\text{sen}}) / \text{Nu}_{\text{mat}} \times 100\% \quad (1)$$

If the mass-based leaf nutrient resorption efficiency was originally estimated (NRE_m), as done in earlier literature (Vergutz et al., 2012; Du et al., 2020), the mass loss correction factor (MLCF) was used to calculate concentration-based leaf nutrient resorption efficiency by means of eq. 8:

$$\text{NuRE} = (1 - (1 - \text{NuRE}_m) \times \text{MLCF}) \times 100\% \quad (2)$$

where *MLCF* is 0.780 for evergreen broadleaves, 0.784 for deciduous broadleaves, 0.745 for conifers, 0.640 for forbs and for 0.713 graminoids, respectively (Vergutz *et al.*, 2012; Du *et al.*, 2020).’ (Page 20, lines 467-475). Notably, these equations of nutrient resorption efficiency are widely accepted and used in other literatures (Vergutz *et al.*, 2012; Du *et al.*, 2020).

*[Comment 20] **Line 311-312**:* How did you address potential biases introduced by log-transforming the data? Were any alternative methods considered?

[Response] Following the reviewer’s comments, we conducted generalized linear mixed-effects models without data-transformation to eliminate potential biases introduced by log-transformation (Bolker *et al.*, 2009). In these models, we set soil mineralization variables as the fixed factors and the replicates nested with site as random factors. These additional analyses showed insignificant relationships between plant N and P resorption efficiency ratio and soil N, P mineralization rates (Figure R3a, b). However, plant N and P resorption efficiency ratio exhibited a significant negative relationship with the ratio of soil N and P mineralization rates (Figure R3c). Nevertheless, we have reorganized this part (Pages 10-11, lines 231-266) and removed this Figure into the Supplementary materials in the revised MS (Page 12, lines 109-117 in the supplementary materials). Thanks for your understanding!

Figure R2. Relationships of leaf N:P resorption efficiency with soil nutrient supply across Tibetan alpine grasslands. **a** Leaf N to P resorption efficiency ratio with topsoil N mineralization rate. **b** Leaf N to P resorption efficiency ratio with topsoil P mineralization rate. **c** Leaf N to P resorption efficiency ratio with topsoil N to P mineralization rate ratio. A significant relationship is shown by a solid line. Error bars denote SE of mean at each site ($n = 3$). Statistics (R^2 and P value) are shown for the generalized linear mixed-effects models.

[Comment 21] **Line 313:** What was the sample size for the independent-samples t -tests, and how did you ensure it was adequate for the comparisons made?

[Response] Combining this comment by this reviewer with the second reviewer’s comment (*graminoid is the major plant functional type in Tibetan Plateau and authors should compare their NRE and PRE with those of global herbs.*), **we re-compiled a more comprehensive global database, and compared the efficiency of nutrient resorption** in plants across Tibetan alpine ecosystem grasslands with those in graminoids and forbs worldwide. In this updated database, **83 and 65 observations** were encompassed for N, P resorption efficiencies in graminoids, and **43 and 27 measurements** in forbs, respectively.

To ensure the adequacy of sample size for the independent-samples t-tests, we conducted power analyses with the ‘pwr’ package in R software 4.3.1 (Cohen, 1988). The results showed that, for leaf N resorption efficiency, the power values were **0.96** (Type I error = 0.05) for the comparison between plants in our study region and graminoids worldwide, and **0.91** for the comparison between plants on the Tibetan Plateau and forbs worldwide. For leaf P resorption efficiency, the power values were **0.92** (Type I error = 0.05) for the comparison with graminoids worldwide, and **0.84** for the comparison with forbs worldwide (Table R2). Given that practitioners usually consider a power value of 0.8 on the basis of the ratio of Type II (β) to Type I error (α) (Cohen, 1988). **This to say, sample sizes used in this study were adequate for these independent-samples t tests.** We have clearly stated this point in the revised MS (Page 21, lines 485-487).

Table R2. Power analyses of the independent-samples t tests for leaf nutrient resorption efficiency between Tibetan alpine grasslands and global herbs.

Growth type	Nutrient resorption	Sample size from Tibetan permafrost region	Sample size from global datasets	α value (type I error)	Cohen's d	Power value
Graminoid	N resorption	30	83	0.05	0.15	0.96
	P resorption	30	65	0.05	0.95	0.92
Forb	N resorption	30	43	0.05	0.35	0.91
	P resorption	30	27	0.05	1.32	0.84

*[Comment 22] **Line 319-320**:* The relationship between senesced and mature leaf nutrient concentrations is crucial. Could you provide additional context or references to support the interpretation of slopes greater than 1 in this context?

[Response] Following the reviewer's suggestion, **we provided additional context and references to support this interpretation in the revised MS as follows:** 'To verify the presence of the nutrient concentration control strategy, we analyzed the relationships between \log_{10} -transformed senesced and mature leaf nutrient concentrations. Specifically, a conceptual model (eq. 2), proposed by Kobe et al. (2005), determines the relationship between senesced and mature leaf nutrient concentrations:

$$Nu_{sen} = a \times Nu_{mat}^b \quad (3)$$

After converting with \log_{10} transformation, a linear regression is shown between $\log_{10}(Nu_{sen})$ and $\log_{10}(Nu_{mat})$ as:

$$\text{Log}(Nu_{sen}) = \text{Log}(a) + b\text{Log}(Nu_{mat}) \quad (4)$$

By combining eq. 1 and eq. 3, leaf nutrient resorption efficiency can be expressed as:

$$NuRE = (1 - aNu_{gr}^{b-1}) \times 100\% \quad (5)$$

Thus, if $b > 1$, nutrient resorption efficiency decreases with increasing mature leaf nutrient concentration (i.e. nutrient concentration control). In contrast, if $b \leq 1$, plant nutrient resorption does not adhere to the nutrient concentration control strategy (Kobe et al., 2005; Sun et al., 2023).' (Pages 15-16, lines 357-371).

Thanks again for this reviewer's insightful and professional review. These comments inspired us to have a deeper thinking on the related methods, data analyses and mechanism explanation behind our results, and thus guided us to conduct a thorough revision of the original MS. To address these insightful comments, **we added the detailed descriptions about the related methods, conducted the additional analyses to assess the rationale of statistical methods and discussed the potential reasons for the finding that plant P resorption is significantly higher than N**

resorption. By doing so, we feel that the revised MS has been greatly improved and expect that the reviewer will be satisfied with the revised manuscript. Thank you!

Responses to Reviewer #2

[Comment 1] Yang and colleagues conducted a sampling campaign along a 1,100 km transect across Tibetan alpine permafrost grassland to explore the N and P resorption efficiency of the grass at the community level. They reported that the mean plant N and P resorption (efficiency), i.e., NRE and PRE, were at the upper end of global synthesis range, and that plant P resorption (efficiency) is higher than N resorption efficiency (75.1%P vs. 58.7%N), which suggested a severe P limitation relative to N on plant growth in the Tibetan alpine permafrost grassland. Their field work is very impressive considering the arduousness of fieldwork on the Qinghai Tibet Plateau. I believe the conclusion about the N and P nutrient status are generally correct. Especially they used data from the in-situ measurements of soil nutrient mineralization to further validate that these permafrost grasslands were more limited by P rather than N.

[Response] Many thanks for the reviewer's positive and insightful comments on our manuscript! These comments listed below inspired us to have a deeper thinking on these issues, and thus guided us to conduct a thorough revision of the original MS. Detailed modifications please see our responses to the following comments. Hopefully, you will be satisfied with these revisions.

[Comment 2] However, the major results are not surprising. Herbaceous plants usually show higher N and P resorption efficiency than woody plants. According to the global results reported by Vergutz et al (2012) (which the authors has cited), the N and P resorption efficiency of herbaceous plants (globally were respectively 67.5~73.3% (NRE) and 79.0~83.3% (PRE) for graminoid, which is the major plant functional type in Tibetan Plateau; or respectively 61.1~73.4% (NRE) and 64.0~77.4% (PRE) for global forbs. All the results of global herbs are higher than Yang's results in this study, and $PRE > NRE$ for global herbs also indicated that herbaceous plants are more limited by P than N. Actually, this is one of the major conclusions in our

current study too. However, it is strangely that the authors compared their NRE and PRE of herbaceous plants on the Qinghai Tibet Plateau in this paper with those of all plant species worldwide (Figure 2). This comparison is clearly unreasonable, leading the author to the wrong conclusion: the NRE and PRE of herbaceous plants in permafrost on the Qinghai Tibet Plateau were higher than the global average. Moreover, the NRE and PRE were OVERESTIMATED in Yang's study because they did not compensate for the leaf mass loss, caused by the strong winds frequently occurs on Tibetan Plateau. Considering this overestimation, their results are even lower than the global results for herbaceous plants reported by Vergutz et al (2012).

[Response] Very good comment! Briefly, the reviewer's comments could be summarized as the following two issues: (1) can plant nutrient resorption efficiency be higher or lower than that reported in this study if considering the mass loss? (2) is plant nutrient resorption efficiency across Tibetan alpine permafrost region at the upper end of global graminoid and forb plants?

Issue 1: Can leaf nutrient resorption efficiency be higher or lower than that reported in this study if considering the mass loss?

As mentioned by this reviewer, mass loss is an important factor which affects the mass-based nutrient resorption efficiency. To account for this effect, the mass loss correction factor (MLCF) was defined as the ratio of the dry mass of senesced leaves and the dry mass of green leaves (< 1). When considering the mass loss effect, mass-based nutrient resorption efficiency is calculated as^{2,7}:

$$NRE_m = (1 - N_{sen} / N_{mat} \times MLCF) \times 100\% \quad (1)$$

By contrast, when ignoring the mass loss effect, the concentration-based nutrient resorption efficiency, frequently used in the literature, is calculated as⁷:

$$NRE_c = (N_{mat} - N_{sen}) / N_{mat} \times 100\% \quad (2)$$

Based on these above two equations, we can find that mass-based nutrient resorption efficiency should be higher than concentration-based nutrient resorption efficiency. Considering that concentration-based nutrient resorption efficiency was represented in

our study, **plant N and P resorption efficiency could be expected to be higher, rather than lower, than that reported in our study if considering the mass loss.**

Issue 2: Is leaf nutrient resorption efficiency across Tibetan alpine permafrost region at the upper end of global forbs and graminoids?

As mentioned by this reviewer, Vergutz et al. (2012) reported that global N and P resorption efficiency were 67.5~73.3% and 79.0~83.3% for global graminoids, and 61.1~73.4% and 64.0~77.4% for global forbs, respectively². **However, we would like to mention that the mass loss was corrected in Vergutz et al. (2012), but not in our study.** To ensure comparability, we calculated nutrient resorption efficiencies in leaves across Tibetan alpine permafrost region on a mass basis (with mass loss correction) used the above-mentioned eq. 1. Community-level MLCF was determined by weighting mass loss correction factor of graminoids and forbs with their corresponding proportion of cover degree at each site. MLCF was 0.640 and 0.713 for forbs and graminoids respectively (Vergutz *et al.*, 2012; Du *et al.*, 2020). Our results showed that mean leaf N and P resorption efficiencies were **70.1 ± 1.0%** and **82.6 ± 1.3%**, respectively. Of them, **leaf N resorption efficiency was at the middle range, while P resorption efficiency was at the upper end of that reported by Vergutz et al. (2012).**

To further verify the above point, we re-compiled a more comprehensive global database including both concentration and mass, and compared the nutrient resorption efficiency in our study with those in forbs and graminoids worldwide. Our results showed that both concentration- and mass-based leaf N resorption efficiencies were approximately equivalent to these of global forbs and graminoids, and both concentration- and mass-based leaf P resorption efficiencies were significantly higher than these of global forbs and graminoids (Fig. 2). **These comparisons supported the conclusion of high leaf P resorption efficiencies across alpine permafrost ecosystems.** We have compared leaf nutrient resorption efficiency in herbaceous

plants between Tibetan alpine permafrost region and global terrestrial ecosystems (Page 6, lines 133-137), and updated the Figure 2 in the revised MS (Page 29, lines 680-689). Combining the first reviewer’s comment (*the analysis presented is appropriate in the global data synthesis section.*), we retained the comparison of leaf nutrient resorption between Tibetan alpine grasslands and globally in the revised MS (Page 9, lines 83-91 in the supplementary materials). Nevertheless, if the reviewer insists on his/her opinion (*this comparison is clearly unreasonable*), we will delete this comparison in next round of revision. Thanks for your understanding!

Figure R3. Comparisons of the concentration- (a) and mass-based (b) leaf nitrogen and phosphorus resorption efficiencies in plants from the Tibetan alpine permafrost region with those in global forbs and graminoids. Data are represented as the means \pm SE. Different letters (independent-samples t tests, $P < 0.05$, lowercase letters for N

and capital letters for P).

Taken together, we are very grateful to the reviewer for the insightful comments on our manuscript! These comments inspired us to have a deeper thinking on the calculations and comparisons of nutrient resorption efficiency. To address these insightful comments, **we re-compiled a more comprehensive global database including both concentration- and mass-based nutrient resorption. Based on these databases, we compared the concentration- and mass-based nutrient resorption efficiency in our study with those in graminoids and forbs worldwide.** By doing so, we feel that our conclusion becomes more convinced, and the revised manuscript has been greatly improved. We expected that the reviewer will be satisfied with the revised manuscript. Thank you!

References

- Bates, D., Mächler, M., Bolker, B. & Walker, S. Fitting Linear Mixed-Effects Models Using lme4. *Stat Comput* **arXiv:1406**, 133-199 (2014).
- Bolker, B.M. *et al.* Generalized linear mixed models: a practical guide for ecology and evolution. *Trends Ecol Evol* **24**, 127-135 (2009).
- Bremner, J.M. & Black, C.A. *Methods of soils analysis: Part 2. Chemical and microbiological properties*. (American Society of Agronomy, Soil Science Society of America, 1965).
- Cohen, J. *Statistical Power Analysis for the Behavioral Sciences* 2nd ed. (Routledge Press, New York, 1988).
- Deng, M. *et al.* Ecosystem scale trade-off in nitrogen acquisition pathways. *Nat Ecol Evol* **2**, 1724-1734 (2018).
- Dolezal, J., Jandova, V., Macek, M. & Liancourt, P. Contrasting biomass allocation responses across ontogeny and stress gradients reveal plant adaptations to drought and cold. *Funct Ecol* **35**, 32-42 (2021).
- Du, E. *et al.* Global patterns of terrestrial nitrogen and phosphorus limitation. *Nat*

- Geosci* **13**, 221-226 (2020).
- Güsewell, S. Nutrient resorption of wetland graminoids is related to the type of nutrient limitation. *Funct Ecol* **19**, 344-354 (2010).
- Hans, L. *et al.* Root structure and functioning for efficient acquisition of phosphorus: Matching morphological and physiological traits. *Ann bot* **4**, 693-713 (2006).
- Hart, S.C., Stark, J.M., Davidson, E.A. & Firestone, M.K. *Methods of soil analysis: Nitrogen mineralization, immobilization, and nitrification*. Chapter 42. (American Society of Agronomy, Soil Science Society of America, 1994).
- Kobe, R. K., Lepczyk, C. A. & Iyer, M. Resorption efficiency decreases with increasing green leaf nutrients in a global data set. *Ecology* **86**, 2780-2792 (2005).
- Lambers, H., Raven, J.A., Shaver, G.R. & Smith, S.E. Plant nutrient-acquisition strategies change with soil age. *Trends Ecol Evol* **23**, 95-103 (2008).
- Lin, Y.M. *et al.* Nutrient conservation strategies of a mangrove species *Rhizophora stylosa* under nutrient limitation. *Plant Soil* **326**, 469-479 (2010).
- Parfitt, R.L. The availability of P from phosphate-goethite bridging complexes. Desorption and uptake by ryegrass. *Plant Soil* **53**, 55-65 (1979).
- Raven, J.A., Lambers, H., Smith, S.E. & Westoby, M. Costs of acquiring phosphorus by vascular land plants: patterns and implications for plant coexistence. *New Phytol* **217**, 1420-1427 (2018).
- Risch, A.C. *et al.* Soil net nitrogen mineralisation across global grasslands. *Nat Commun* **10**, 4981 (2019).
- Güsewell, S., & Schroth, M. How functional is a trait? Phosphorus mobilization through root exudates differs little between *Carex* species with and without specialized dauciform roots. *New Phytol* **215**, 1438-1450 (2017).
- Shen, M.G. *et al.* Plant phenology changes and drivers on the Qinghai-Tibetan Plateau. *Nat Rev Earth Env* **3**, 633-651 (2022).
- Sun, X. *et al.* Widespread controls of leaf nutrient resorption by nutrient limitation and stoichiometry. *Funct Ecol* **37**, 1653-1662 (2023).

- Vergutz, L. *et al.* Global resorption efficiencies and concentrations of carbon and nutrients in leaves of terrestrial plants. *Ecoll Monogr* **82**, 205-220 (2012).
- Wanek, W. *et al.* A novel isotope pool dilution approach to quantify gross rates of key abiotic and biological processes in the soil phosphorus cycle. *Biogeosciences* **16**, 3047–3068 (2019).
- Wen, Z., White, P.J., Shen, J. & Lambers, H. Linking root exudation to belowground economic traits for resource acquisition. *New Phytol* **233**, 1620-1635 (2021).
- Yan, B. *et al.* Plants adapted to nutrient limitation allocate less biomass into stems in an arid-hot grassland. *New Phytol* **211**, 1232-1240, (2016).
- Yang, G. *et al.* Characteristics of methane emissions from alpine thermokarst lakes on the Tibetan Plateau. *Nat Commun* **14**, 3121 (2023).

Responses to Reviewer #1

[Comment 1] This is my second time reviewing this manuscript, and I thoroughly enjoyed reading it. I am impressed by the substantial improvements made to the approach and the robustness of the analyses. The authors have done an excellent job, and this study provides valuable insights into how climate change may influence the dynamics and functions of permafrost—an ecosystem of critical importance that remains understudied. The introduction has been significantly improved, and I am pleased with the clarity and logical flow of the language. The authors have addressed my previous questions and comments comprehensively, and I am satisfied with their thoughtful and thorough responses.

[Response] We appreciate this reviewer for his/her recognition on our revised manuscript.

Responses to Reviewer #2

[Comment 1] The authors have made major revisions according to the previous comments. The quality of the paper has been greatly improved. I have no further suggestions, and recommend accepting this manuscript for publication in NC.

[Response] We thank this reviewer for his/her positive feedback on our revised manuscript.

Reviewer #1 attachment:

Below are comments that follow more or less the order in the paper, not order of importance.

1. **Ln 1***: The title is inappropriate; "adaptation" is not an appropriate term. I suggest "alpine permafrost ecosystem."
2. **Ln 26-29***: I suggest changing it to "projections hold that larger plant nitrogen (N) resorption relative to phosphorus (P)" to make it more natural.
3. **Ln 28***: What's the cold region? It should be "permafrost ecosystem."
4. **Ln 30***: "Two times sampling campaign" should be simplified to "two sampling campaigns."
5. **Ln 32-33***: P resorption rate? What's the statistical confidence? Or what's the standard error? And why significant? What's the p-value?
6. **Ln 32***: Changing "is significantly higher relative" to "was significantly higher relative to" maintains tense consistency.
7. **Ln 45-47***: Not clear; suggest changing to "Therefore, a better understanding and more precise estimation of plant nutrient resorption are essential for accurately evaluating plant nutrient status and predicting terrestrial C dynamics at regional and global scales."
8. **Ln 45-47***: Using "approximately" and "around" instead of "~" is more formal and professional in the introduction section.
9. **Ln 59-61***: Adding "have" clarifies the tense. "Long-standing paradigm" is more accurate than "long-term standing paradigm."
10. **Ln 62-63***: "Suggested" is more commonly used in scientific literature than "advocated." Changing "at nutrient-limited region" to "in nutrient-limited regions" corrects the preposition and pluralizes "region" for consistency.
11. **Ln 94***: Capitalizing key words in the title aligns with standard scientific publication formats. I suggest "High Plant Nutrient Resorption in Alpine Ecosystem across Tibetan Plateau."
12. **Ln 95-96***: "Based on" is more concise than "On the basis of." The phrase "geographical patterns" is clearer and more precise.
13. **Ln 99-100***: Simplifying the sentence structure improves readability and flow. I suggest "Plant N resorption efficiency ranged from 37.9% to 72.3%, averaging $58.7 \pm 1.5\%$, while plant P resorption efficiency varied from 44.4% to 87.3%, with a mean value of $75.1 \pm 1.8\%$."
14. **Ln 101-102***: Removing "of them" and "respectively" simplifies the sentence without losing meaning.
15. **Ln 109***: "Ulteriorly" is not a commonly used word and can be replaced with "subsequently" for clarity. Correct the spelling of "varying" and "respectively."
16. **Ln 112-113***: "Nutrient resorption by the plant" is more concise than "resorption of nutrient by the plant."

17. ****Ln 120-121****: Removing "plant" for conciseness because "N and P resorption efficiencies" clearly refers to plant nutrient resorption.
18. ****Ln 127-128****: Adding "than N resorption efficiency" clarifies the comparison being made.
19. ****Ln 129-130****: Splitting the sentence and clarifying the conditions for better readability and grammar. I suggest "Generally, plants have roughly balanced N and P resorption efficiency under balanced growth conditions. However, if there is N or P limitation, plants will resorb a greater proportion of the limited nutrient from senesced leaves."
20. ****Ln 134-144****: Adjust the sentence for clarity and ease of understanding. Correct "in-suit" to "in-situ" and adjust the preposition for better clarity.
21. ****Ln 146-148****: Break the sentence into two for clarity and add "However" to clearly indicate the contrast.
22. ****Ln 155-157****: Remove "Actually" for conciseness and ensure consistent tense.
23. ****Ln 158-160****: "Overall" is a more formal transition than "All in all." "Challenge" is more appropriate than "are against," and "long-standing paradigm" is clearer.
24. ****Ln 167-169****: Split the sentence for clarity and flow, and remove unnecessary words for conciseness. I suggest "Our results revealed that senesced leaf N concentrations did not show any significant relationship with mature leaf N concentrations ($P = 0.13$; Figure 3a). However, senesced leaf P concentration was significantly correlated with mature leaf P concentration ($P < 0.001$), with a slope of 1.1 (Figure 3b)."
25. ****Ln 181-184****: Correct "Plant may" to "Plants may" for subject-verb agreement, and simplify for clarity and conciseness.
26. The summary section provides a clear conclusion of the study. However, several areas need refinement for clarity, coherence, and conciseness, such as adding "campaigns" for grammatical accuracy and "a" before "poorly-studied" for clarity in Ln 200, replacing "blind-spot" with "understudied" for a more formal tone, and simplifying the sentence for clarity in Ln 202. Change "evidences" to "evidence" for correct usage, and simplify the sentence for clarity in Ln 212.
27. ****Ln 238-239****: Simplify the sentence for clarity and conciseness. Change "two times' field campaigns" to "two field campaigns."
28. ****Ln 238-239****: What were the specific criteria for selecting the 30 sampling sites? Were there any particular environmental or vegetation characteristics considered?
29. ****Ln 254-256****: Could you provide more details on the calibration and accuracy of the elemental analyzer and spectrophotometer used in the study?
30. Was any consideration given to potential seasonal variations in nutrient resorption rates beyond the specified sampling periods? If so, how were they addressed?
31. ****Ln 273-275****: Why were three plastic tubes chosen per site, and how does this number ensure the representativeness of the data?

32. ****Ln 293****: How were the initial corresponding values in soils before incubation measured, and what steps were taken to ensure their accuracy?
33. ****Line 299-300****: Could you specify the criteria used to select the published datasets included in the global data synthesis? Were there any exclusion criteria?
34. ****Line 303****: How did you ensure the exclusion of data from wetlands, aquatic, and agricultural ecosystems was consistent across all sources?
35. ****Line 305-307****: Could you provide more detail on the equations used to calculate global plant N and P resorption efficiencies? Are these equations widely accepted in the field, or were any modifications made?
36. ****Line 311-312****: How did you address potential biases introduced by log-transforming the data? Were any alternative methods considered?
37. ****Line 313****: What was the sample size for the independent-samples t-tests, and how did you ensure it was adequate for the comparisons made?
38. ****Line 319-320****: The relationship between senesced and mature leaf nutrient concentrations is crucial. Could you provide additional context or references to support the interpretation of slopes greater than 1 in this context?